# A Murine Model of High Dietary Histamine Intake: Impact on Histamine Contents and Release in Neural and Extraneural Tissues

**DOI:** 10.3390/nu17111851

**Published:** 2025-05-29

**Authors:** Annette Kuhn, Jana Schramm, Birgit Vogler, Mária Dux, Fernando de Mora, Karl Messlinger

**Affiliations:** 1Institute of Physiology and Pathophysiology, Friedrich-Alexander-University, D-91054 Erlangen, Germany; annette.kuhn@fau.de (A.K.); jana.schramm@fau.de (J.S.); birgit.vogler@fau.de (B.V.); 2Department of Physiology, University of Szeged, H-6720 Szeged, Hungary; dux.maria@med.u-szeged.hu; 3Department of Pharmacology, Therapeutics and Toxicology, Universidad Autónoma de Barcelona, 08193 Bellaterra, Spain; fernando.demora@uab.cat

**Keywords:** histamine intolerance, dietary histamine, plasma histamine, histamine release, mast cells, trigeminal system, mouse

## Abstract

**Background:** Histamine intolerance, a disorder due to impaired degradation of dietary histamine, is frequently associated with headaches, but the underlying pathophysiology is largely unknown; the sensitization of meningeal afferents appears likely. We approached this issue by examining histamine concentrations in different tissues and meningeal histamine release in a new mouse model of high-histamine diets. **Methods:** C57BL/6 mice of both sexes were fed with diets containing 3 or 9 g/kg histamine and compared to control groups. After 10–30 days, the histamine concentration was determined in plasma, samples of homogenized ileum, trigeminal ganglia, spinal medulla, and cerebellum using an ELISA. The histamine release from mast cells in the dura mater stimulated with compound 48/80 was also examined. **Results:** Animals supplied with high dietary histamine showed normal behavior and no signs of suffering. Compared with the controls, the histamine concentration was significantly higher in plasma and ileum of mice fed with 3 g/kg, highest in animals fed with 9 g/kg histamine. In addition, this group of animals showed also higher histamine concentrations in the trigeminal ganglion. The histamine release from the dura mater in mice supplied with 3 g/kg histamine was not significantly different to control animals, but the relative increase in stimulated release was lower in male animals of the high histamine group. **Conclusions:** High dietary histamine increases histamine levels in blood plasma and the gut, whereas the histamine content of neural tissues is not significantly influenced. The lowered stimulated release in animals subjected to high dietary histamine may indicate compensatory mechanisms.

## 1. Introduction

The oral intake of normal or even low levels of histamine present in certain foods may cause detrimental effects in a subpopulation of individuals with a low expression of diamine oxidase (DAO), the main enzyme involved in exogenous histamine degradation [1]. Histamine intolerance (HIT), also referred to as enteral histaminosis or sensitivity to dietary histamine, is therefore defined as a disorder associated with an impaired ability to metabolize ingested histamine [2,3]. Such deficiency commonly affects the gastrointestinal, dermal, and/or the respiratory systems [4,5,6], but it has also been associated with headaches and triggering migraine attacks [7,8,9]. These and other authors claim that diet histamine-to-migraine attack association is linked to the amount of histamine present in certain foods. So-called HIT is, therefore, directly linked to the generation of neural symptoms in certain individuals. Although we have recently attempted to propose a mechanism involved in exogenous histamine-driven migraine [10], the pathophysiology of the detrimental consequences of HIT in any organ/system has been scarcely studied and, therefore, is largely unknown. Further insights into the relevance and underlying mechanism of diet-induced, histamine-associated migraine may help identify a potential role for exogenous diamine oxidase (DAO) or alternative therapeutic interventions in affected patients. Among other question marks, there is a complete lack of understanding of the kinetics throughout the body of undegraded dietary histamine, whether taken in large quantities or at normal levels by DAO-deficient patients. In spite of its short half-life according to in vivo animal models, some authors suggested that gut histamine may diffuse into the systemic circulation, as experiments with histamine application to rodent gastric mucosa have shown [11]. However, local gastrointestinal effects of exogenous histamine may be able to cause pathological conditions associated with HIT; namely, for instance, the activation of afferent neurons in the gut that may ultimately foster headache and migraine induction by releasing sensitizing mediator molecules in or even distantly from the gastrointestinal system. The lack of models mimicking the impact of real-life fluctuations of dietary histamine has contributed to preventing science from advancing in the knowledge of the mechanisms leading to symptoms in HIT in a diversity of systems.

Based on the exogenous histamine-to-migraine scenario, this study aimed to evaluate the effect of a high-histamine diet on histamine levels in neural and extraneural tissues and histamine release from meningeal mast cells, in a novel murine model of high dietary histamine intake. The approach to this analysis included several variations in high-histamine dietary feeding and analyzing histamine concentrations in the tissues, since experiments and measurements of this kind are largely lacking so far.

## 2. Materials and Methods

### 2.1. Animals and Ethics

111 young adult C57BL/6J mice with a body weight (mean ± SD) of 27.5 ± 3.7 g (males) and 21.1 ± 1.5 g (females) were purchased from Charles River (Sulzfeld, Germany), individually ear-marked, and held in a 12 h day–night-cycle in groups of 3 animals of same sex in standard type III cages (according to ETS 123) with wood chip bedding enriched with tissue paper and kidney dishes for hiding. The animals were fed with standard raising food pellets (V1124-000, Ssniff GmbH, Soest, Germany) and water ad libitum for about one week for acclimatization. Animal housing and all experiments were carried out according to the German guidelines and regulations of the care and treatment of laboratory animals and the European Communities Council Directive of 24 November 1986 (86/609/EEC), amended 22 September 2010 (2010/63/EU).

### 2.2. Low and High Dietary Histamine Fed Mice

The experimental approach, including diets provided per mouse group, composition, and timelines, is summarized in Figure 1 and Table 1. No animal was excluded from the assignment. After acclimatization, four cohorts of 12 animals each (sexes balanced) were matched according to their sex and weight and randomly allocated to the groups shown in Table 1. The animals were continuously supplied either with Altromin C1072 containing no additional histamine and low amounts of 4.675 mg/kg feed weight histidine (low-histamine control group) or with Altromin C1072 with additional 3 g/kg histamine for 3–4 weeks (i.e., 22–23, 23–24, or 31–32 days) or for a short period of 3 days (high-histamine test groups). The diet from Altromin GmbH (Lage, Germany) was chosen according to previous information from the literature [12]. The other two cohorts of 12 animals each were continuously supplied either with standard raising food V1124-000 (produced by Ssniff GmbH, Soest, Germany) containing 0.58% histidine or with standard raising food plus 9 g/kg histamine dihydrochloride (ultra-high-histamine test group) for 19–20 or 21–22 days. As an advantage, the animals had already been raised with this raising food at Charles River, so that in the control group with low histamine, no change in diet was necessary. Table 1 shows the breakdown of experimental groups.

Noticeably, histamine content in foods varies widely from less than 0.5 mg/kg in fruits or cereals, for example, to more than 500 mg/kg in ripe (aged) cheese, non-fresh fish, or certain wines [2]. The contents reached in our proposed diets study (histamine concentrations of 3 and 9 g/kg food) were well over what may be found in the high-end scale, but it was limited by the maximum that the companies could add to the pellets.

The histamine content per diet is reflected in Table 1. The precise composition of Altromin C1072 and Ssniff is available as Appendix A. Every day between 11 a.m. and 1 p.m., the consumed food in each cage and the weight of each animal were determined until the end of the feeding periods. For that, animals were handled and inspected for any uncommon conditions or behavior.

### 2.3. Tissue Sample Preparation

After the feeding period, animals were deeply anesthetized with isoflurane, thoracotomized, and exsanguinized by collecting the blood from the heart with a syringe containing EDTA (Merck, Darmstadt, Germany) for preventing coagulation. The blood was immediately centrifuged, and the supernatant plasma was collected and deep-frozen. The head was separated and skinned. The occipital bone was carefully removed, and the whole medulla, together with the cerebellum, was separated and immediately frozen. In some samples, about 1 cm of the ileum was dissected and frozen. The trigeminal ganglia of both hemi-skulls were dissected immediately after the release experiments (see below), pooled, and frozen. After defrosting, the tissue samples were dipped in a filter paper to remove adhering fluid, weighed, heated in 20 µL 0.01% perchloric acid and 180 µL EIA buffer at 95 °C, then homogenized using a custom-made homogenizer, again heated in the above solution at 95 °C, centrifuged, and neutralized with 1.5 M NaOH. Then, 10 µL of the supernatant was taken off and diluted with 990 µL EIA buffer to determine the histamine content (see below). EIA (enzyme-linked immunoassay) buffer was from Bertin Pharma/SPIbio, Montigny le Bretonneux, France. 

### 2.4. Histamine Release Experiments

The head of the animal was divided into the sagittal plane, and the hemispheres were removed without touching the dura mater lining the skull. The hemi-skulls were washed in neutral buffer solution (synthetic interstitial fluid, SIF), mounted on a water bath at 37 °C, and filled two times with 200 µL SIF for 5 min and then with a solution of compound 48/80 in SIF for 5 min each to induce histamine release. The composition of SIF was (in mM): 108 NaCl, 3.48 KCl, 3.5 MgSO_4_, 26 NaHCO_3_, 11.7 NaH_2_PO_4_, 1.5 CaCl_2_, 9.6 Na-gluconate, 5.55 glucose, and 7.6 sucrose; pH 7.4. From each solution, samples of 100 µL were taken off and supplemented with 25 µL concentrated EIA buffer (Bertin Pharma/SPIbio, Montigny le Bretonneux, France). All fluid samples were frozen for the histamine measurements, which were usually performed within one week.

### 2.5. Histamine Concentration Assessment

The samples containing histamine were measured using an ELISA (Bertin Pharma/SPIbio, Montigny le Bretonneux, France) based on the competition between the histamine in the samples and the tracer, acetylcholine (Ach) linked to histamine. Wells that are precoated with a specific antibody against mouse histamine are used, which binds free and linked histamine. The reaction of Ach with Ellman’s reagent forms a yellow compound, the intensity of which is inversely proportional to the amount of free histamine in the samples. First, the histamine standards were prepared with assay buffer (Bertin Pharma) to a concentration of 500 nM histamine and step-wise diluted with SIF to concentrations in a descending order, ranging from 50 to 0.39 nM histamine. The plasma samples were diluted 1:20 with SIF, as well as tissue samples 1:50 with SIF. Lower dilutions were probed but turned out to be inferior for this ELISA. Then, 200 µL of each standard and sample was mixed with 50 µL derivatization buffer (Bertin Pharma) and 20 µL derivatization reagent. The derivatization reagent (Bertin Pharma), which increases the affinity of histamine to the antibody and the sensitivity of the assay, was reconstituted with N-N-dimethyl formamide; DMSO turned out to be unsuitable. From each of the standard and sample mixtures, 100 µL were pipetted into the anti-histamine antibody-coated wells of a plate, which had been carefully pre-washed with wash buffer solution (Bertin Pharma) and complemented with 100 µL histamine acetylcholine esterase tracer (Bertin Pharma). The plates were incubated for 24 h at 4 °C, then emptied and rinsed 5 times with wash buffer solution, and dried. Finally, 200 µL of Ellman’s reagent was added to each well, with the plate covered with aluminum foil, and incubated at room temperature in the dark on an orbital shaker. After 60 min, the plate was analyzed in a spectrophotometer, reading the absorbance of the yellow color at a wavelength of 405 nm. The histamine concentration in the samples was determined by comparison with a sigmoid standard curve based on the graded series of histamine solutions. The detection limit of the assay is 0.5 nM (=0.056 ng/mL), and its cross-reactivity to histamine and serotonin is <0.01% according to the manufacturer. The inter-assay coefficient of variation for plasma samples is 15.8%; for tissue samples, it is 5.2%. The intra-assay coefficient of variation for plasma samples is 15.1%; for tissue samples, it is 7.07%, according to the manufacturer.

### 2.6. Toluidine Blue Staining of Dura Mater

C57BL/6 mice (4 males and 3 females) fed with either Ssniff enriched with 9 g/kg histamine or Ssniff without additional histamine (control) were sacrificed and prepared as described for histamine release. The cranial dura mater was carefully dissected from the skull halves, washed in SIF, and stored in phosphate-buffered saline at 4 °C (PBS). For staining, the dura halves were incubated in 0.1% toluidine blue (Merck, Darmstadt, Germany) dissolved in saline at room temperature for 2 min, then dipped in SIF, differentiated in 70% ethanol for 20 s, washed in PBS, stretched on a microscope slide, and embedded. Micrographs of mast cells were taken with an LSM 780 microscope (Carl Zeiss MicroImaging GmbH, Jena, Germany) mounted on an inverted Axio Observer Z1 using the light microscope mode with a 20× dry objective lens and a 40× water immersion lens (numerical apertures 0.8 and 0.95).

### 2.7. Calculations and Statistics

In blood plasma, the histamine content was expressed as ng/mL. The measured histamine content of homogenized tissues was related to the weight of the tissue samples so that the histamine concentration (ng/mg) could be calculated. Likewise, in addition to the measured concentration in the buffer solution, the histamine released from the cranial dura mater was related to the body weight of the individual animals as a rough estimation of the head size and hence the area of the dura. The consumed food and the body weight of animals between the high-histamine and the control groups were compared with the Student *t*-test for independent samples. Concentration data in the tissues of different groups were compared with factorial analysis of variance (ANOVA) with the independent factors “histamine diet” and “sex”. Release data were tested with repeated measures ANOVA for the three consecutive steps in histamine release with the same independent factors. In case of significant differences, the Tukey honest significant difference (HSD) test was applied to specify the significances. A probability level of *p* < 0.05 was regarded as statistically significant. Data are shown as means ± standard deviation (SD) or standard error of means (SEM) for better differentiation.

## 3. Results

### 3.1. Condition, Food Consumption, and Changes in Body Weight

A precise description of the experimental results, as well as an interpretation, is provided below.

#### 3.1.1. Animals Supplied with Altromin for 3–4 Weeks

During the whole feeding period, the animals in all groups stayed healthy and showed normal behavior and no signs of suffering. In the first three days (i.e., day 1–3) of switching from standard food to a diet from Altromin with low or with high histamine (3 g/kg), the consumption dropped in all groups and stayed on a lower level (Figure 2A). According to the higher body weight, male mice of both groups consumed more food than female mice. Animals fed with a high-histamine diet consumed less food than mice fed with a low-histamine diet (*t*-test for independent samples, males *p* < 0.0001, females *p* < 0.05), which aligned slowly within two weeks of feeding. After two weeks of feeding, food consumption tended to become irregular, possibly because of external influences in the animal house.

During 24 days of feeding, mice of all groups gained weight, males more than females (Figure 2B). Mice fed with a low-histamine diet tended to gain more weight than mice fed with a high-histamine diet during the whole feeding period (*t*-test for independent samples, *p* = 0.065 for males, *p* = 0.095 for females).

#### 3.1.2. Animals Supplied with Ssniff for About 3 Weeks

Animals in both groups showed healthy conditions and normal behaviors during the whole feeding period. On the first day (day 1) after switching from standard food to a diet with ultra-high histamine (9 g/kg), the consumption dropped both in male and female animals, whereas this was not the case in control animals continuously supplied with standard food without histamine (Figure 3A). According to the higher body weight, male mice of both groups consumed more food than female mice, but the low-histamine groups did not significantly differ from the ultra-high-histamine groups (*t*-test for independent samples, *p* = 0.12 in males and 0.54 in females).

During 20 days of feeding with low- or ultra-high-histamine diet, mice of all groups gained weight, males more than females (Figure 3B). Mice of both sexes fed with low histamine had more weight on average than mice fed with ultra-high-histamine diet during the feeding period (*t*-test for independent samples, *p* < 0.001 for males and females).

### 3.2. Histamine Content of Blood Plasma, Ileum, Trigeminal Ganglia, and Cerebellum

The concentration of histamine was determined in blood plasma as pg/mL and in the ileum (one of the samples got lost), the trigeminal ganglia, medulla oblongata, and cerebellum as ng/mg tissue. Because the results of the three cohorts of animals supplied with Altromin without or with 3 g/kg histamine in the diet for 3–4 weeks (i.e., 22–23, 23–24, or 31–32 days) were statistically not different, the data were pooled (Figure 4A). Factorial ANOVA showed significant differences in histamine concentration between low- and high-histamine diet groups in the plasma (*F*_1,28_ = 9.45, *p* < 0.005) and the ileum (*F*_1,28_ = 20.35, *p* < 0.0005) but not in the trigeminal ganglia and the cerebellum. Since there was also no sex difference in any of the factors, data of both sexes were not differentiated. The data of the cohort of animals fed with Altromin for only three days (Figure 4B) were similar, and only the ileum showed significant differences between high and low dietary histamine (*F*_1,8_ = 13.13, *p* < 0.05). Animals fed with Ssniff without or with 9 g/kg histamine (Figure 4C) showed significant differences in histamine concentration between low- and ultra-high-histamine diet groups in plasma (factorial ANOVA, *F*_1,20_ = 10.38, *p* < 0.005), ileum (*F*_1,19_ = 21.17, *p* < 0.000) and additionally in the trigeminal ganglia (*F*_1,20_ = 4.41, *p* < 0.05) but not in the medulla (*F*_1,20_ = 0.40, *p* = 0.53) and the cerebellum (*F*_1,20_ = 0.29, *p* = 0.60). In none of the tissues, ANOVA indicated a significant sex difference; thus, data of both sexes are pooled in the image.

### 3.3. Histamine Release from Mouse Dura Mater

In mice fed with Altromin without or with 3 g/kg histamine for 23–32 days, the basal histamine release from the cranial dura mater in both hemisected heads and the histamine release provoked by compound 48/80 was determined and normalized to the body weight of each animal in 18 males and 14 females (Figure 5A). In addition to the normalized histamine release, stimulated release data were calculated relative to the mean of the two basal release values (Figure 5B). There was a general difference between the sexes with higher values in females (repeated-measure ANOVA, *F*_1,58_ = 23.75, *p* < 0.0005). Therefore, the sexes are differentially displayed in Figure 5. Neither in basal nor in compound 48/80-stimulated histamine release were significant differences observed between animals supplied with a high-histamine diet or control food. While a tendency towards higher basal values (at 5 min) in animals fed with high-histamine diet is visible Figure 5A), the stimulated release tended to be lower in high-histamine diet animals, which is best expressed when data are normalized to the basal release (Figure 5B), where the difference in stimulated release between low- and high-histamine diet animals is significant in male animals (*, repeated measures ANOVA and Tukey post hoc test, *p* > 0.01).

### 3.4. Toluidine Blue Staining of Mast Cells in Mouse Dura Mater

Mast cells, as the most prominent source of histamine in the cranial dura mater, were visualized in seven animals. After staining with toluidine blue and differentiation with ethanol, the mast cells appeared red-violet on a background of blue meningeal tissues (Figure 6A,B). Mononuclear cells other than mast cells, possibly contributing to the histamine content of the dura mater, were very rarely visible (Figure 6B).

## 4. Discussion

### 4.1. Impact of High Histamine on Animal Conditions

High and ultra-high dietary histamine is obviously well tolerated by wild-type mice that did not show any unusual behaviors or signs of suffering, although we cannot exclude undetected differences in their health condition in comparison to control mice. Expectedly, the body weight was generally lower in females. Both sexes gained weight with either diet, but the growth ratio was slightly higher in males. However, mice fed with a high-histamine diet lagged behind in food consumption and body weight compared to mice fed with a normal diet, so they possibly had an altered or disturbed digestion or metabolism during the feeding period. Another explanation could be a direct influence of the plasma histamine level on the regulation of food intake by hypothalamic mechanisms, where neuronal histamine has been reported to have a suppressive effect through thyreotropin-releasing hormone [13] and leptin [14]. Nevertheless, the body weight gain tended to become equal towards the end of the feeding periods, so the groups were suitable for comparisons.

We did not compare food intake dependent on the two diets, Altromin and Ssniff (both with and without additional histamine), because this was not the focus of our study. Indeed, the average intake was slightly higher in animals fed with Ssniff compared to animals consuming Altromin. Apart from the above-mentioned influence of histamine on the hypothalamic regulation of food intake [13], a small difference in convertible energy of the two diets (Altromin 15.7 MJ/kg, Ssniff 14 MJ/kg) may lead to slightly more intake of Ssniff compared to Altromin. In addition, the histidine content of the Ssniff diet was much higher compared to the Altromin diet. Normal dietary histidine levels have been found to be critical for normal weight gain in mice subjected to a low-protein diet [15].

### 4.2. Histamine Levels in Different Tissues

In order to assess the kinetics/distribution of dietary histamine and/or its impact on endogenous histamine production, histamine levels were measured in plasma and the ileum. Obviously, only a small fraction of the range of minus three potencies of histamine in the diet was resorbed, i.e., 3 or 9 mg/kg histamine in the high- or ultra-high-histamine diet resulted in a histamine concentration of 8 or 12 mg/g in the ileum tissue. Most of the histamine in the gut may be stored in mast cells [16], which are differentiated into three types according to topological functional criteria in rodents [17]. Not surprisingly, in the circulating plasma, histamine was even at four potencies less than in the diet. Nevertheless, under the given conditions, the histamine concentrations measured in plasma and the ileum seemed to reflect the exposure to the substantial difference in dietary histamine intake. In plasma, the histamine levels were more pronounced after a longer feeding period with 3 g/kg histamine (compare Figure 4A,B) and after feeding with ultra-high histamine content of 9 g/kg (compare Figure 4A,C), which may be understood as a matter of equilibrium with the histamine concentration in the gut. Also, the histidine content of the food, which was about one potency higher in Ssniff compared to Altromin diet (see Table 1), may have had some influence on the histamine concentration in plasma when converted by gut bacteria [18]. However, the differences should not be over-interpreted, because the precision of the histamine measurements is limited by a rather high variation in the measurements resulting from the complicated assay and the high inter-assay variability.

The plasma histamine levels measured in our study are not consistent with reports about histamine infusion in human test persons, where concentrations of more than 2 ng/mL elicited symptoms like an increase in heart rate and blood pressure, flush, and headache [19]. Although we did not record such clinical data, mice did not show any unusual signs even at ultra-high-histamine diets. Indeed, histamine baseline concentrations of 20–100 ng/mL are regarded as normal in rodents [20]. In addition, different analysis methods (radioenzyme assay in the human study vs. ELISA in our study) may produce different results, so these studies cannot be compared.

### 4.3. Histamine Uptake into Tissues

Considering the role of histamine as an inflammatory mediator for nociceptive afferents and as a neurotransmitter in the central nervous system, we measured the histamine concentration in the peripheral (i.e., the trigeminal ganglion) and the central trigeminal system (i.e., the medulla spinalis containing the spinal trigeminal nucleus), and in the cerebellum as a central neural structure. From our measurements, the histamine concentration in the diet and in plasma did not seem to have a major influence on the histamine uptake into these neural tissues. The medulla and cerebellum showed the lowest histamine concentration, independent of the dietary histamine. This is conceivable, since these tissues are protected by the blood–brain barrier and are not directly accessible by plasma histamine; histamine is rather formed within the brain from histidine, which is transported through the blood–brain barrier [21], although mast cells may also penetrate the blood–brain barrier under pathological conditions [22]. Previous measurements revealed very different concentrations in the brain, e.g., 0.06 ng/g cerebellar tissue from rabbit [23] or 4.4 ng/mg in rat cerebellum [24], again, as a matter of different assays making quantitative comparisons between the studies virtually impossible. In the latter study, histamine concentrations were measured in different brain areas after a high dietary histidine load (1 g/kg twice daily for 1 week). There was a significant increase in histamine concentration compared to control animals in the cortex and the hypothalamus, but not in the cerebellum [24].

In the present study, the trigeminal ganglion, which is outside of the blood–brain barrier [25], showed significantly higher levels of histamine compared to the cerebellum and the medullar brainstem, but we do not know in which cellular compartment it is present. A very preliminary look into trigeminal ganglion slices showed very few mononuclear cells that may store histamine, but it would require immunohistochemical techniques with antibodies directed against mouse histamine to confirm this.

### 4.4. Mast Cells in the Dura Mater

In the dura mater, it is fairly clear that mast cells are the most prominent source of histamine, although other mononuclear cells such as basophils cannot be excluded (see Figure 6). Dural mast cells are mainly arranged along arterial blood vessels and have a close spatial association with meningeal nerve fibers, as has long been demonstrated [26,27]. The number of mast cells was rather variable, and their density seemed not to be different between animals fed with a high-histamine diet compared to the controls, although we did not perform precise countings. With the inclusion of these limitations, we assume that the histamine diet has no significant impact on the formation or migration of mast cells into the dura mater, which is consistent with the lack of significant differences in basal histamine release.

### 4.5. Histamine Release from the Dura Mater

The histamine release seemed to be marginally influenced by dietary histamine. Normalized to the body weight of animals as an approximate measure of the dural area, the basal histamine release tended to be higher in animals fed with high histamine, while the release stimulated by the mast cell degranulator compound 48/80, an approved method to degranulate mast cells effectively [28], tended to be lower in high-histamine animals. Normalized to the basal histamine release, the stimulated histamine release in mice fed with a high-histamine diet (3 g/kg) was significantly lower in males. The stimulated histamine release may, therefore, rather depend on a higher sensitivity or instability of mast cells, particularly in the male dura mater.

### 4.6. Limitations and Future Perspectives

One limitation of the study is that it is virtually impossible to feed a diet without any histidine, so we have zero control over the histamine diet. In a preliminary study, we fed some mice an amino acid diet without histamine and histidine, which led to a critical loss of body weight within a few days, so we had to stop this experiment. On the other end of the scale, even with the very high concentration of histamine, only part of it is resorbed in the gut and shows up in the blood plasma. An even more severe confounder is the enzymatic activity of the organisms breaking down histamine with the action of diamino oxidase (DAO). This is also the limitation in terms of transition to the clinic, since histamine intolerance is mostly due to a failure or lack of DAO activity. Therefore, as a next step to using our model of histamine release and concentration in different tissues, DAO knockout mice should be used to constitute a situation similar to the clinical problem. Attempting to understand whether the lower stimulated histamine release reflects compensatory mechanisms, possibly receptor desensitization or feedback inhibition, or both, may be an interesting follow-up endeavor. Finally, it may be interesting to incorporate nociceptive behavior evaluations through the study of neural activation indicators (e.g., c-Fos) or CGRP to enhance the translational relevance of the study.

## 5. Conclusions

We present an in vivo model for low and high dietary histamine intake and histamine measurements in various tissues that may contribute to an advance in the knowledge of histamine intolerance mechanisms. We conclude that histamine diffusion or transport from the gut to the plasma seems to be likely, based on the present data showing higher histamine levels in high-histamine-fed mice. The stimulated histamine release from mast cells of the dura mater seems not to be significantly influenced by dietary histamine but rather by sex-dependent factors. Mast cell degranulation may contribute to the sensitization of meningeal afferents in headache generation and migraine attacks independently of dietary histamine levels.

## Figures and Tables

**Figure 1 nutrients-17-01851-f001:**
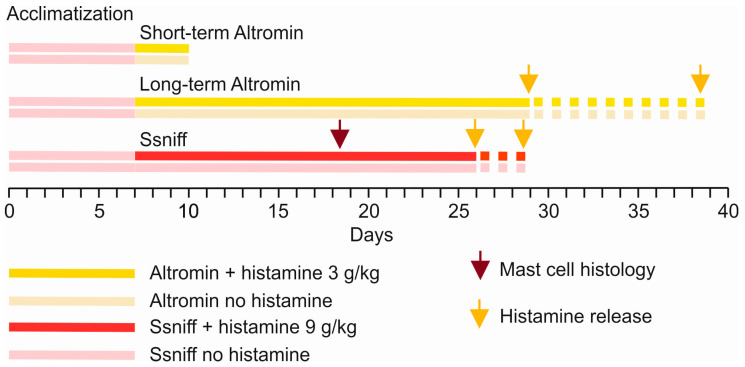
Timelines for the mice acclimatization and consumption of the diets, either with no histamine or extra histamine content. Two cohorts were fed with standard raising food (Ssniff) with or without additional histamine. Arrows point to the time of mast cell histology or histamine release/content analyses.

**Figure 2 nutrients-17-01851-f002:**
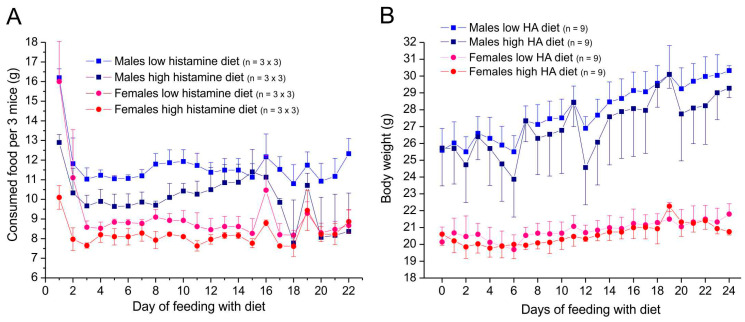
Food intake (**A**) and body weight (**B**) of animals supplied with Altromin. Data in A are means from 3 cages with 3 mice each, error bars mean ± SEM (only +SEM shown in upper curves). Data in B are means of individual animals, error bars mean ± SD (only +SD or −SD shown).

**Figure 3 nutrients-17-01851-f003:**
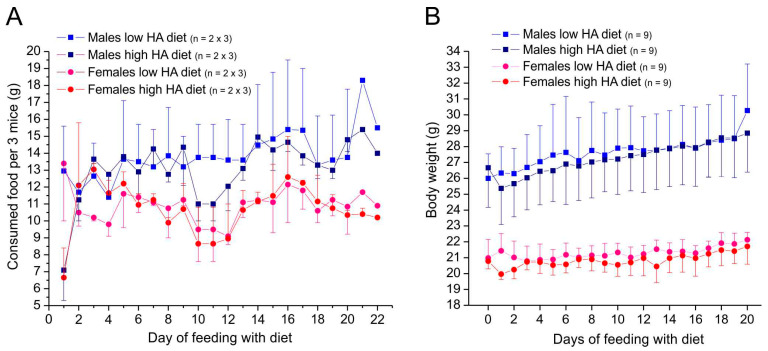
Food intake (**A**) and body weight (**B**) of animals supplied with Ssniff. Data in A are means from 2 cages with 3 mice each, error bars mean ± SEM (only +SEM or −SEM shown). Data in B are means of individual animals, error bars mean ± SD (only +SD or −SD shown).

**Figure 4 nutrients-17-01851-f004:**
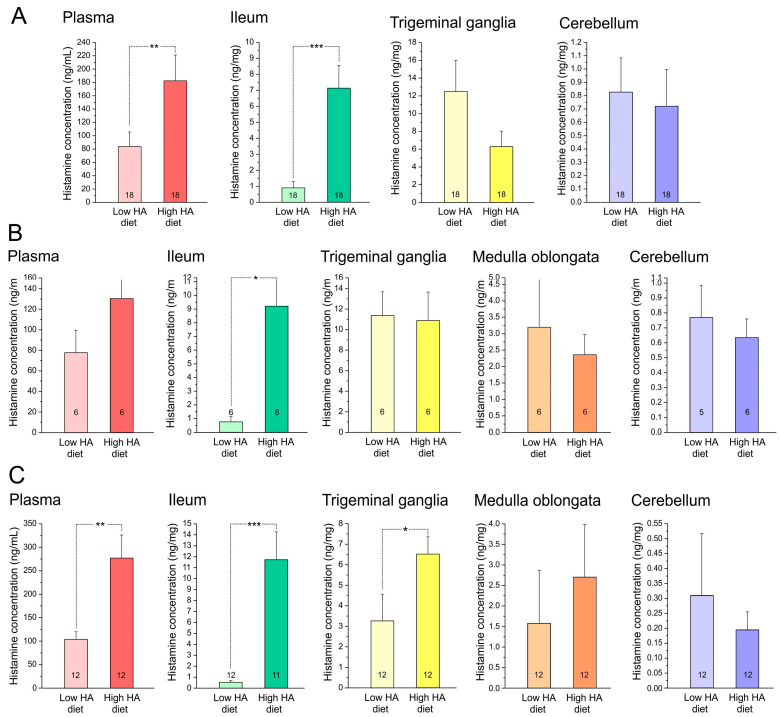
Histamine concentration in mouse plasma (ng/mL), homogenized ileum, trigeminal ganglia, medulla oblongata, and cerebellum (ng/mg) in animals fed with Altromin with 3 g/kg histamine for 22–32 days (**A**) or for 3 days (**B**); or in animals fed with Ssniff with 9 g/kg histamine for 19–22 days (**C**), compared with respective control groups fed with low-histamine diet. Values are means ± SEM (only +SEM shown) and compared with factorial ANOVA extended by the Tukey post hoc test; ***, *p* < 0.0005; **, *p* < 0.005; * *p* < 0.05. Numbers of animals are shown in the bars.

**Figure 5 nutrients-17-01851-f005:**
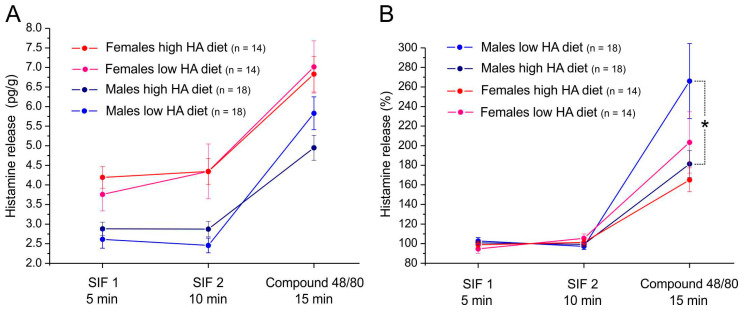
Histamine release from the cranial dura mater in the mouse hemisected skull preparation, normalized to the body weight of the animals (**A**) and expressed relative to the basal release, the mean of SIF1 and SIF2 values (**B**). High-histamine diet was 3 g/kg. * *p* < 0.01, repeated measures ANOVA and Tukey post hoc test. Data are means ± SEM.

**Figure 6 nutrients-17-01851-f006:**
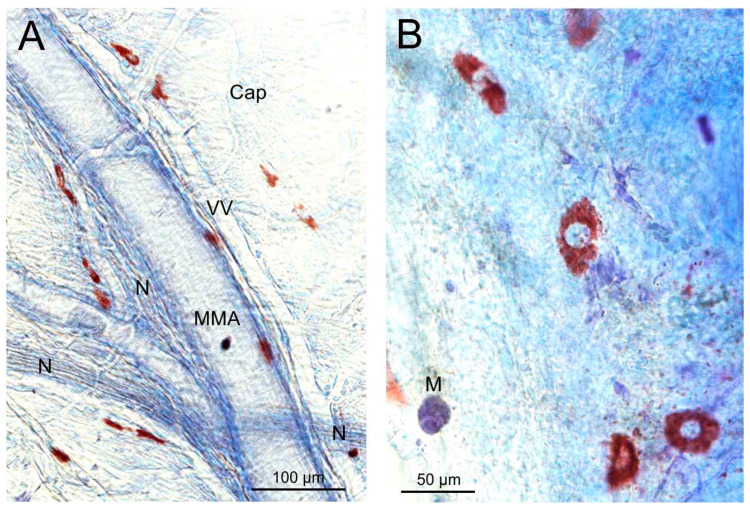
Mast cells in the cranial dura mater of a control animal. They are mostly arranged along the middle meningeal artery (MMA) and second- to third-order arteries, as can be seen in (**A**). At higher magnification, the granulated structure of the mast cells is visible (**B**). Cap, capillary connecting precapillary artery and venous vessel (VV) running along the MMA; M, mononuclear cell; N, meningeal nerve bundles.

**Table 1 nutrients-17-01851-t001:** Diets and experimental groups.

Diet Group	Histidine Content (mg/kg)	Histamine Content (g/kg)	Feeding Period (Days)	Group Size (*n*) Males (M) Females (F)	Experiments Histamine (HA)
Altromin C1072 Control	4.675	0	22–32	9 M/9 F	HA content
				9 M/7 F	HA release
			3	3 M/3 F	HA content
Altromin C1072 Test	4.675	3	22–32	9 M/9 F	HA content
				9 M/7 F	HA release
			3	3 M/3 F	HA content
Ssniff Control	5800	0	19–22	6 M/6 F	HA content
Ssniff Test	5800	9	19–22	6 M/6 F	HA content
Ssniff Control	5800	0	10–12	2 M/1 F	Mast cells
Ssniff Test	5800	9	10–12	2 M/2 F	Mast cells

## Data Availability

Further data underlying the presentation of this study are available upon request from the corresponding author.

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
