# Peer review of "A Murine Model of High Dietary Histamine Intake: Impact on Histamine Contents and Release in Neural and Extraneural Tissues"

_nutrients, 2025, doi:10.3390/nu17111851_

Round 1

Reviewer 1 Report

Comments and Suggestions for Authors

The authors in their work have taken up an important topic concerning dietary histamine intake.

The introduction is a proper introduction to the topic. The objective is correctly formulated, and the research has been planned in such a way as to enable an answer to the problem contained in the objective.

In the materials and methods, I propose to add a Scheme of the research model - what was done in which order and on which group, although the authors use tables, it seems to me that such a scheme would additionally allow the reader to follow the research better.

I have no objections to the presentation of the results; everything is consistent here, the figures are legible and appropriately marked.

The authors should add the Limitations of the study.
They should also add information about what their article contributes and what the next research steps should be.

I believe that the list of 4 abbreviations is unnecessary

Author Response

The authors in their work have taken up an important topic concerning dietary histamine intake.

The introduction is a proper introduction to the topic. The objective is correctly formulated, and the research has been planned in such a way as to enable an answer to the problem contained in the objective.

Response: We thank the reviewer for appreciating our study. English has been slightly modified.

In the materials and methods, I propose to add a Scheme of the research model - what was done in which order and on which group, although the authors use tables, it seems to me that such a scheme would additionally allow the reader to follow the research better.

Response: We have added a scheme showing the different experiments with timeline. We hope that this is what the reviewer asked for (please find it attached). The other figures have been renumbered accordingly.

I have no objections to the presentation of the results; everything is consistent here, the figures are legible and appropriately marked.

The authors should add the Limitations of the study.

Response: A section “Limitations and future perspectives of the study” has been added. The text follows: “One limitation of our study is that it is virtually impossible to feed a diet without any histidine, so that we have no zero control of the histamine diet. In a preliminary study, we fed some mice with an amino acid diet without histamine and histidine, which led to a critical loss of body weight within few day, so that we had to stop this experiment. On the other end of the scale, even with the very high concentration of histamine, only part of it is resorbed in the gut and shows up in the blood plasma. An even more severe confounder is the enzymatic activity of the organisms breaking down histamine with the action of diamino oxidase (DAO). This is also the limitation in terms of transition to the clinic, since histamine intolerance is mostly due to a failure or lack of DAO activity. Therefore, as a next step to use our model of histamine release and concentration in different tissues, DAO knockout mice should be used to constitute a situation similar to the clinical problem.”

They should also add information about what their article contributes and what the next research steps should be.

Response: Please see above.

I believe that the list of 4 abbreviations is unnecessary

Response: We have removed the abbreviation list.

Reviewer 2 Report

Comments and Suggestions for Authors

Title Suggestion:
Effects of High Histamine Diets on Tissue Histamine Levels and Meningeal Mast Cell Release in a Mouse Model

Some sentences are long and would benefit from clearer segmentation for readability.
The objectives could be stated more explicitly, e.g., “This study aimed to evaluate…”
Units (3 or 9 g/kg) should specify whether they refer to feed weight.

The model's relevance to human histamine intolerance symptoms (e.g., headache models) could be better contextualized.
The choice of histamine concentrations (3 g/kg and 9 g/kg) needs brief justification.

Duration of intervention is mentioned (10–30 days), but it's unclear if results differ by time point.
Details on ELISA sensitivity, quantification limits, and normalization methods would strengthen methodological transparency.
Was the histamine release assay from dura mater standardized per number of mast cells?

The statement “the histamine uptake into neural tissues is not significantly influenced” might benefit from specifying p-values or effect sizes.
The phrase “lowered stimulated release may indicate compensatory mechanisms” should suggest whether this could relate to receptor desensitization or feedback inhibition.

Consider suggesting implications for future research, e.g., whether prolonged exposure could lead to sensitization rather than compensation.
A brief statement on translational potential to human histamine intolerance would add impact.

Author Response

Title Suggestion:
Effects of High Histamine Diets on Tissue Histamine Levels and Meningeal Mast Cell Release in a Mouse Model

Response: Although we acknowledge and thank for the reviewer’s suggestion, we have only slightly changed it. It is relevant to acknowledge right from scratch that this is in fact a new model to study high dietary histamine. Such a model has never been proposed: hence our request to mention it ahead. We agreed to add a reference to histamine released as per the reviewer suggestion. We hope that the Reviewer finds it suitable as it now stands: “A murine model of high dietary histamine intake: impact on histamine contents and release in neural and extraneural tissues”.

Some sentences are long and would benefit from clearer segmentation for readability.
The objectives could be stated more explicitly, e.g., “This study aimed to evaluate…”
Units (3 or 9 g/kg) should specify whether they refer to feed weight.

Response: We have reworded the manuscript accordingly.

The model's relevance to human histamine intolerance symptoms (e.g., headache models) could be better contextualized.

Response: We have attempted to better convey the message.

The choice of histamine concentrations (3 g/kg and 9 g/kg) needs brief justification.

Response: We wanted to increase the histamine load as far as possible. The histamine concentrations of 3 and 9 g/kg food was the maximum the companies could add to the pellets.

Duration of intervention is mentioned (10–30 days), but it's unclear if results differ by time point.

Response: The histamine concentration in mouse tissues was first analysed in animals fed with Altromin with 3 g/kg histamine for 22-32 days to simulate a chronic state of histamine overload. The differences in waiting days was necessary due to technical reasons but there was no statistical difference within this group. To examine if a short-term feeding would yield similar results, we added a small group with 3 days Altromin food. We recognized that the plasma histamine concentration was somewhat higher after long-term feeding (see Figure 4). For the even higher histamine concentration, which was only possible to realize with Ssniff food, we used an intermediate waiting time of 19-22 days, which was close to the long-term feeding with Altromin, again without statistical difference within this group.

Details on ELISA sensitivity, quantification limits, and normalization methods would strengthen methodological transparency.
Was the histamine release assay from dura mater standardized per number of mast cells?

Response: The histamine release was not standardized to the number of mast cells. Mast cells can only be counted microscopically after preparation of the dura from the skull but it hardly possible to do this without lesions and loss of parts of the dura.

The statement “the histamine uptake into neural tissues is not significantly influenced” might benefit from specifying p-values or effect sizes.

Response: The author is right in pointing out this mistake. We have actually no data for the uptake. Sentence has been changed accordingly.

The phrase “lowered stimulated release may indicate compensatory mechanisms” should suggest whether this could relate to receptor desensitization or feedback inhibition.

Response: The reviewer points at an interesting potential explanation. Given that we would like not to keep it too speculative and given the fact we have not attempted to sort out the reason, we believe that it is better not to further discuss it. We have mentioned it in a new section “Limitations and future perspectives”.

Consider suggesting implications for future research, e.g., whether prolonged exposure could lead to sensitization rather than compensation.
A brief statement on translational potential to human histamine intolerance would add impact.

Response: Histamine intolerance is mostly caused by a loss of function of the main histamine degrading enzyme, diamine oxidase (DAO). Thus overload of histamine is not the reason for histamine intolerance but the high histamine dose may aggravate symptoms in this disease. As suggested by the 1st reviewer, we have added a paragraph stating the clinical link to histamine intolerance.

Reviewer 3 Report

Comments and Suggestions for Authors

The manuscript seeks to associate dietary histamine with migraine via trigeminal and meningeal pathways, although the relationship remains conjectural. There is an absence of functional or behavioral data corroborating the sensitization of trigeminal afferents or the manifestation of migraine-like symptoms in mice. Incorporating nociceptive behavior evaluations, neural activation indicators (e.g., c-Fos), or CGRP quantifications would significantly enhance the translational relevance.

The research primarily examines histamine concentrations in tissues without evaluating any functional or physiological effects, such as alterations in pain sensitivity, neuroinflammation, or behavioral reactions. This complicates the ability to ascertain the clinical significance of the observed histamine distribution.

The research focuses on histamine intolerance; nevertheless, the activity of diamine oxidase (DAO), the principal mechanism for histamine breakdown, is neither quantified nor experimentally addressed. This is a significant deficiency, as variations in histamine levels may result from modified metabolism instead of uptake or release.

The histamine quantification measurements exhibit significant variability. The authors should address ELISA limitations by providing inter-assay coefficients of variation, validating sensitivity, and considering confirmation of results by LC-MS/MS or another reliable approach. Enhanced statistical reporting, including effect sizes and confidence intervals, is also necessary.

The data indicate sex-specific effects on histamine release, although they are not sufficiently analyzed. A comprehensive examination and discourse on possible hormonal or cellular mechanisms (e.g., the influence of sex hormones on mast cell function) would augment the manuscript's originality.

The publication designates mast cells as the primary source of histamine; nevertheless, it fails to measure mast cell populations, degranulation status, or release kinetics beyond stimulation with compound 48/80. This constrains mechanistic understanding. Supplementary staining (e.g., tryptase) or quantification would be advantageous.

Numerous previous research have investigated dietary histamine and its systemic impacts. The uniqueness assertion should be more clearly delineated by highlighting what is genuinely innovative (e.g., measurement in trigeminal ganglia, utilization of ultra-high histamine diets) and how this contributes to forthcoming research on HIT or neuroimmune interactions.

The disparities in dietary formulation (Altromin vs. Ssniff) and histidine concentration between the control and experimental groups introduce confounding variables. These factors are neither regulated nor standardized, and they may affect histamine production. A distinct control utilizing the same basic meals devoid of histamine supplementation is essential for accurate interpretation.

The study fails to address the ethical or translational ramifications of administering ultra-high amounts of histamine to mice. Were the dosages physiologically pertinent to human exposure in HIT? Comparing the findings to established human histamine exposures might provide perspective.

Certain figures are challenging to read, such as the histamine concentration data in Figure 3 and the release data in Figure 4. The utilization of absolute values and standard error of the mean without individual data points constrains clarity. Incorporating scatter plots or box plots with statistical comments would enhance transparency.

Author Response

The manuscript seeks to associate dietary histamine with migraine via trigeminal and meningeal pathways, although the relationship remains conjectural. There is an absence of functional or behavioral data corroborating the sensitization of trigeminal afferents or the manifestation of migraine-like symptoms in mice. Incorporating nociceptive behavior evaluations, neural activation indicators (e.g., c-Fos), or CGRP quantifications would significantly enhance the translational relevance.

Response: We thank the reviewer for his/her suggestions to include functional and behavioral data. We have carefully monitored the behaviour of animals and found not any difference to the control mice. So we could not decide about a test which might be specific for histamine overload. Regarding c-Fos staining, this is a very good idea, however, this would require fixation of the tissue, which was not possible, because we need unfixed tissue for homogenization and quantitative histamine measurements. CGRP release is indeed what we have done in another study, the date of which will be published in another manuscript. We have mentioned this in a new section “Limitations and future perspectives”.

The research primarily examines histamine concentrations in tissues without evaluating any functional or physiological effects, such as alterations in pain sensitivity, neuroinflammation, or behavioral reactions. This complicates the ability to ascertain the clinical significance of the observed histamine distribution.

Response: The report is a first attempt to see if high histamine supply causes any changes in histamine content of tissues. Therefore, we wanted to address this pioneering character of our study by naming it in the title as “murine model of high dietary histamine intake”. Follow up studies should certainly focus on functional and physiological effects, but the novel of such a murine model is, in our view, of high interest for the reader of Nutrients. The availability of the model will ensure further understanding of the clinical significance.   

The research focuses on histamine intolerance; nevertheless, the activity of diamine oxidase (DAO), the principal mechanism for histamine breakdown, is neither quantified nor experimentally addressed. This is a significant deficiency, as variations in histamine levels may result from modified metabolism instead of uptake or release.

Response: The reviewer is right in the statement that the current research on dietary histamine is histamine intolerance with its main reason, the failure or loss of diamine oxidase (DAO). This could be addressed with DAO knockout mice, a project which we had discussed to start with, however, as a first step we wanted just to find out whether high histamine supply induces any changes in histamine content of neural and non-neural tissues. The reviewer proposed alternative studies that will certainly be of interest as follow up. The importance of our model is that they can now be conducted in vivo in such high diet histamine mice. 

The histamine quantification measurements exhibit significant variability. The authors should address ELISA limitations by providing inter-assay coefficients of variation, validating sensitivity, and considering confirmation of results by LC-MS/MS or another reliable approach. Enhanced statistical reporting, including effect sizes and confidence intervals, is also necessary.

Response: Histamine ELISA is a rather validated method for histamine quantification (we have no access to LC-MS/MS). As for the Reviewer suggestion we have now added coefficients of variation (Inter-assay coefficient of variation for plasma samples 15.8%, for tissue samples 5.2% / Intra-assay coefficient of variation for plasma samples 15.1%, for tissue samples 7.07%). In any case the info can be found in the reference kit.

The data indicate sex-specific effects on histamine release, although they are not sufficiently analyzed. A comprehensive examination and discourse on possible hormonal or cellular mechanisms (e.g., the influence of sex hormones on mast cell function) would augment the manuscript's originality.

Response: We appreciate the reviewers’s question of sex-specific differences. The n numbers did not allow to sufficiently differentiate male and female results. We think that discussing this issue is beyond the scope of this first study but may indeed interesting for a follow-up.

The publication designates mast cells as the primary source of histamine; nevertheless, it fails to measure mast cell populations, degranulation status, or release kinetics beyond stimulation with compound 48/80. This constrains mechanistic understanding. Supplementary staining (e.g., tryptase) or quantification would be advantageous.

Response: Again, these studies are beyond the scope of the project, in spite of being of interest.

Numerous previous research have investigated dietary histamine and its systemic impacts. The uniqueness assertion should be more clearly delineated by highlighting what is genuinely innovative (e.g., measurement in trigeminal ganglia, utilization of ultra-high histamine diets) and how this contributes to forthcoming research on HIT or neuroimmune interactions.

Response: We introduced a novel animal model for high histamine diet and combined it with the established method measuring the release of histamine. Activation and degranulation of meningeal mast cells may play a central role in augmenting neurogenic inflammatory reaction in the dura mater that may sensitize trigeminal nociceptors. This is all stated throughout the manuscript, and, in our view, justified the spreading of the data through the scientific community.

The disparities in dietary formulation (Altromin vs. Ssniff) and histidine concentration between the control and experimental groups introduce confounding variables. These factors are neither regulated nor standardized, and they may affect histamine production. A distinct control utilizing the same basic meals devoid of histamine supplementation is essential for accurate interpretation.

Response: The control groups were fed with standard food without additional histamine, both in the Altromin and the Ssniff groups. Other basic foods are not available. Further we explain it in the material and methods where a scheme has been included.

The study fails to address the ethical or translational ramifications of administering ultra-high amounts of histamine to mice. Were the dosages physiologically pertinent to human exposure in HIT? Comparing the findings to established human histamine exposures might provide perspective.

Response: We have started with a small cohort of animals to see if there are behavioural responses or other signs due to the high histamine load but have not any unusual change. Following this experience, we have increased the number of animals but continued to carefully monitor them. The loss of weight during the first days of switching to high histamine food was within the tolerance fixed in the animal application. The ethical issues have been addressed in the methods.

Certain figures are challenging to read, such as the histamine concentration data in Figure 3 and the release data in Figure 4. The utilization of absolute values and standard error of the mean without individual data points constrains clarity. Incorporating scatter plots or box plots with statistical comments would enhance transparency.

Response: We appreciate the opinion of the reviewer but cannot really follow his/her concerns. Showing mean with standard deviation or standard error of the mean is absolute praxis for data, which are normally distributed, while median with box plots is usually used if this is not the case. In our experience, scatter and box plots would be more confusing than means and errors.

Round 2

Reviewer 3 Report

Comments and Suggestions for Authors

Accepted